# Changes in UK ophthalmology surgical training: analysis of cumulative surgical experience 2009–2015

Jeremy Hoffman,[1,2] Fiona Spencer,[3,4] Daniel Ezra,[1,2] Alexander C Day[1,2]

[1]NIHR Biomedical Research Centre, Moorfields Eye Hospital NHS Foundation Trust, London, UK
[2]UCL Institute of Ophthalmology, London, UK
[3]The Manchester Royal Eye Hospital, Manchester, UK
[4]The Royal College of Ophthalmologists, London, UK

**Correspondence to**
Dr Jeremy Hoffman;
j.hoffman@ucl.ac.uk

## ABSTRACT

**Objective** To investigate changes in the patterns of cumulative surgical experience for ophthalmologists in the UK following the introduction of a new national training scheme.

**Design** Retrospective review of all surgical training records submitted to the UK Royal College of Ophthalmologists by trainees for the award of Certificate of Completion of Training (CCT) for the period 2009–2015.

**Setting** Secondary level care, UK.

**Participants** 539 trainees achieving CCT over the 7-year study period.

**Interventions** Higher specialist training or ophthalmology specialist training.

**Outcome measures** Number of CCT awards by years and procedures performed for cataract surgery, strabismus, corneal grafts, vitreoretinal (VR) procedures, oculoplastics and glaucoma.

**Results** Cataract surgical experience showed little change with median number performed/performed supervised (P/PS) 592, IQR: 472–738; mean: 631. Similarly, the median number of strabismus (P/PS 34), corneal grafts (assisted, 9) and VR procedures (assisted, 34) appeared constant. There was a trend towards increasing surgical numbers for oculoplastics (median 116) and glaucoma (57). Overall case numbers for ophthalmic specialist training (OST) trainees (7-year training programme) were higher than higher surgical training (HST) trainees (4.5-year programme) with the exception of squint (P/PS), corneal grafts (P/PS) and VR cases (P/PS).

**Conclusions** Overall case numbers reported at time of CCT application appear stable or with a marginal trend towards increasing case numbers. HST (4.5-year programme) case numbers do not include those performed before entry to HST, and although case numbers tended to be higher for OST trainees (7-year programme) compared with HST trainees, they were not proportionately so.

## INTRODUCTION

The aim of ophthalmic specialist training (OST) is to produce competent consultant ophthalmologists of the future.[1] The way that this is achieved for both ophthalmology and all other medical and surgical specialties, has evolved rapidly throughout the developed world over the last two decades, changing from a largely time-based apprenticeship system as introduced by William Halstead in 1904 to a

### Strengths and limitations of this study

► Surgical training records were provided for all UK ophthalmology trainees applying for award of Certificate of Completion of Training by the Royal College of Ophthalmologists over a 7-year period. The results are expected to provide generalisable data for future comparisons both nationally and internationally.

► Case numbers were self-reported from trainees' logbooks. However, these are verified at each year of training by a named educational supervisor and the cumulative numbers by the local training programme director.

► Summary numbers do not take into account additional other surgical experience.

► The individual time in training cannot be taken into consideration and may be heterogeneous, particularly for trainees prior to Modernising Medical Careers and those who have had atypical training pathways.

► There may be regional variations in surgical case numbers due to differences in opportunities and training between deaneries that this study does not assess.

competency-based system.[2] This is particularly true in the UK, which has undergone a number of specific regulatory and legislative changes in the last 20 years. The first major change was the 'Calman Report' in 1996 which combined the Registrar and Senior Registrar grades into a 54-month Specialist Registrar (SpR) training programme.[3] The second change was the so-called 'New Deal' for junior doctors, which introduced a banding system from 2001 to pay for out-of-hours work, essentially resulting in a change from an 'on-call' rota to a 'shift' system (although within ophthalmology this has largely remained an on-call system apart from for a few specialist eye hospitals).[4 5] This was followed by the European Working Time Directive (EWTD) in 2004, which saw a statutory enforcement of the maximum number of hours worked, from 58 in 2004 to 48 in 2009, unless employees actively opted out.[6 7] In addition to these

changes in the working hours of junior doctors in training, a shift to a competency-based curriculum was introduced through the Modernising Medical Careers (MMC) initiative.[8 9] Despite this, many specialties, particularly procedural-intense specialties including ophthalmology, have kept a minimum length of training time requirement in addition to the competency-based curriculum.

Within ophthalmology in the UK specifically, there has been a paradigm shift in the way training is provided and the requirements for completion. Prior to the Calman report, training was largely time-based and one completed training after 3 years at senior registrar level. After 'Calmanisation' in 1996–2007, Ophthalmology Basic Surgical Training (BST, while a Senior House Officer (SHO)) was of typical duration 2–4 years, followed by competitive entry to higher surgical training (HST, SpR) which lasted 4.5 years, with exposure to all subspecialties. Typical roles for a BST SHO, as a junior level trainee, included being the 'first' member of the on-call team, seeing patients initially before discussing with a more senior 'second' on-call, often an HST SpR. A BST SHO would also be expected to see fewer patients in an outpatient setting than their HST SpR counterparts, and in surgery would often be assisting their HST SpR colleague or consultant. An HST SpR on the other hand would have more responsibility and would be expected to perform more surgical procedures independently. However, they would still be under their consultant's supervision. Following the introduction of MMC in ophthalmology in 2008, BST and HST were combined to give a 7-year programme: OST is now provided as a 'run-through' scheme from specialty training (ST) year 1 to ST7, the final ST7 year, including the opportunity to spend 6–12 months undertaking a trainee-selected component in a subspecialty. ST1 to ST3 are often considered analogous to the BST years, while ST4 and above to the HST years. However, the distinction since MMC is less, with an emphasis on a gradual, steady increase in competency with increasing time in training. There are also set criteria for each year of training for trainees to complete in order to progress, which includes work-based assessments (WBAs), examinations and feedback from peers. Subspecialty exposure is continuous throughout training, with trainees expected to have achieved a prescribed number of subspecialist curricular requirements by the end of their training. After successful completion of training, specialty trainees receive a Certificate of Completion of Training (CCT, formally Certificate of Completion of Specialist Training). Successful completion requires satisfactory performance at Annual Review of Competence Progression (ARCP) appraisals, which in turn requires evidence of fulfilling curricular requirements including meeting the minimum number of WBAs, evidence of more generic professional capabilities and satisfactory assessment by both clinical and educational supervisors as well as peers. Awarding of CCT also requires successful completion of the FRCOphth exit examination, required since 2001.

There has been widespread concern across all medical specialties that reduced hours and the introduction of

shift patterns have had a detrimental impact on surgical experience.[10 11] There was also concern that with the introduction of new fast-track independent cataract centres there would be a detrimental impact on cataract surgical training.[12] In order to answer the question of how surgical training experience has changed since the early 1990s, Ezra *et al* published data on the cumulative surgical experience of senior registrar and SpR ophthalmology trainees when they reached the end of their training.[13] Interestingly, their results suggested that despite the backdrop of the New Deal, EWTD and MMC, combined with the introduction of independent sector treatment centres and diagnostic treatment centres potentially reducing the number of suitable training cases, cataract surgical experience had stayed constant. There had, however, been a reduction in subspecialist procedures performed.

The aim of this study is to report on any changes in the patterns of cumulative surgical experience following the first cohort of UK ophthalmologists to have completed their training, achieving CCT, since the instigation of MMC with the introduction of the OST programme and the phasing out of HST.

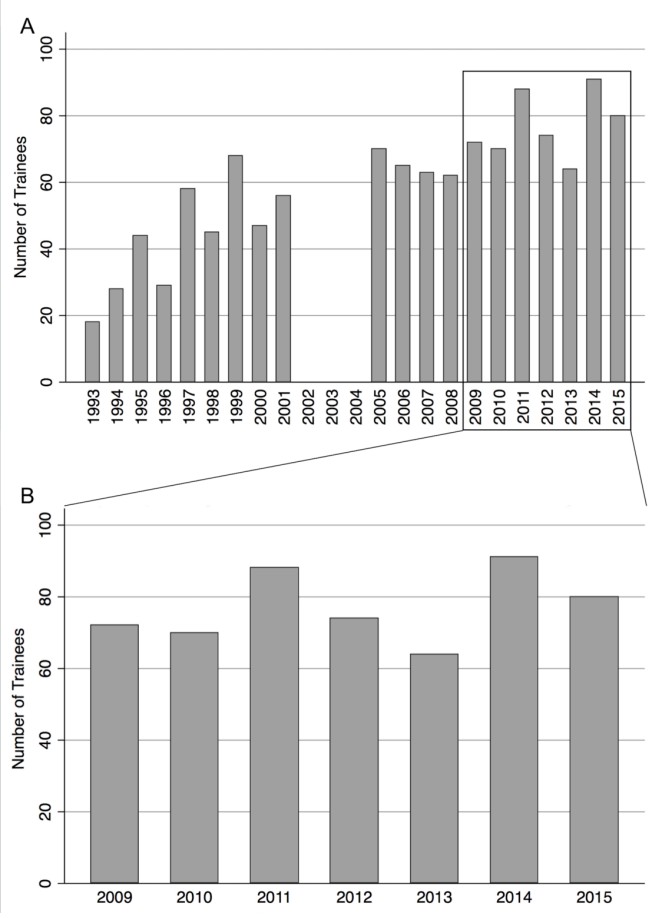

**Figure 1** (A) Frequency histogram of the number of trainees achieving accreditation, Certificate of Completion of Surgical Training (CCST) or Certificate of Completion of Training (CCT). Data from 1993 to 2008 from Ezra *et al*.[13] Note missing data between 2002 and 2004; (B) Frequency histogram of the number of trainees achieving accreditation during our 7-year study period, CCST or CCT.

## METHODS

Anonymous cumulative surgical data were obtained from the Education and Training Department at the Royal College of Ophthalmologists (RCOphth) for every trainee at their time of CCT application for the period 2009–2015 inclusive.

The categories of cumulative surgical experience were the same as that previously reported:[13] cataract surgery, strabismus surgery, oculoplastic surgery, vitre-oretinal (VR) procedures and corneal transplants, and additionally for glaucoma. For VR and corneal surgery, the curricular requirements are to have assisted (A) in procedures, not performed surgery, whereas for cataract, strabismus, glaucoma and oculoplastic procedures, these should have been performed by the trainee and both supervised (PS) and unsupervised (P) cumulative numbers are included, while the assisting totals for these subspecialties were excluded. Although not a curricular requirement, we also included the cumulative total of P and PS cases for VR and corneal graft surgery to allow for comparison to previous data. Comparison data for the periods 1993–2001 and 2005–2008 were obtained from the previous analysis by Ezra et al.[13]

Interquartile ranges (IQRs) for each surgical category were displayed as box and whisker plots using STATA V.14.0. Box and whisker plots demonstrate the quartile spread of the data distribution of cumulative procedures for each of the surgical subspecialties. The central box represents the central two quartiles and the whiskers represent the upper and lower quartiles. Outliers are represented by circles. Analysis is descriptive. No missing data were present as this would have resulted in CCT not being awarded. Statistical comparison between OST and HST groups was performed by Mood's median test to calculate the probability that the null hypothesis is accepted, that is, the medians of the two groups are identical. Statistical comparison of the means of the OST and HST groups was performed by the t-test. No ethical approval was required for this study.

## RESULTS

Data were obtained from the RCOphth on 539 trainees achieving CCT between 2009 and 2015. There were no missing data values for the individual components of surgical experience. The mean number of CCT awards was 77 per year (SD 10, range 64–91). Figure 1A shows data from the current analysis to that previous; figure 1B shows data from our current study.

The cumulative cataract surgical experience from 2009 to 2015 is shown in table 1 and figure 2. The median numbers 'performed' has stayed largely constant. The overall median for the 7-year study period was 592 cases (mean 631, SD 223, range 206–1700) with the lowest annual median of 511 in 2009 to a high of 641 in 2014.

For VR surgery, case numbers 'performed' at the time of CCT remained similar, although there was considerable variation between trainees with some reporting >1000 cases.

Case numbers for 'assisted' remained unchanged (table 1). Figure 3A shows the medians by year of all 'performed' cases with the outliers removed to allow for scaling.

'Performed' cases numbers for corneal graft surgery were similar across the study period, again with skew towards low case numbers similar to VR procedures. The median case number performed ranged from 0 in 2012, 2014 and 2015 to 2 in 2010 (figure 3B, table 1). 'Assisted' case median numbers were 9 per year over the study period (mean 12, SD 10, range 6–89) table 1.

For squint surgery, median number 'performed' at CCT per year again remained constant, with overall study period median of 34 cases (mean 42, SD 34, range 2–304), lowest median in 2009 of 29 and maximum of 37 in 2010 (table 1, figure 4A).

For oculoplastics, there was a trend towards increasing case numbers 'performed' between 2009 and 2015 with low of 94 in 2011 to a high of 152 in 2015. There remain several outliers with much higher numbers, but the trend for truncation in the lower quartiles has stopped. The median number across the 7-year period was 116 (mean 158, SD 133, range 10–1010) (table 1, figure 4B).

The median number of glaucoma procedures 'performed' has increased between 2009 and 2015 from 51 to 72, with an overall median of 57 across the study period (mean 80, SD 63, 17–530) (table 1, figure 5).

Comparisons between HST (4.5-year programme) and OST (7-year programme) trainees showed higher cataract numbers (performed), corneal grafts (assisted), oculoplastic cases (performed), ptosis surgeries (assisted), glaucoma cases (performed), VR cases (assisted) and retinal laser cases (performed) for OST trainees. Squint cases (performed), corneal grafts (performed) and VR cases (performed) showed no significant difference over the study period (table 1).

## DISCUSSION

The mean number of ophthalmology trainees achieving CCT annually by the RCOphth during the study period was 77 (SD 10), with a peak of 91 gaining CCT in 2014. The numbers of ophthalmologists in training have increased since 1993 following the Calman report recommending expansion of the consultant workforce, but there was no significant change in numbers during the current study period.

Cumulative numbers relating to cataract surgery have remained constant over the last two decades at between 500 and 600 cases by the end of training, comparing the current study periods and previous reports by Ezra et al[13] The numbers are maintained despite the pressures of the EWTD and 'Calmanisation' of training. This level of experience compares favourably with our colleagues in the USA, where in 2015–2016 the median number of cataract surgeries performed by residents at the end of training was 173.[14] Indeed, the minimum number of cataract surgeries as defined by the American Council for Graduate Medical Education is set at 86.[15] Intraocular surgical experience

**Table 1** Data showing required surgical case numbers, and cases performed overall and by HST (n=303) and OST (n=236) trainees over the 2009–2015 study period

| Surgery | Current OST minimum number for CCT | Median, IQR (mean, SD), range | HST median, IQR (mean, SD), range | OST median, IQR (mean, SD), range | Difference in means, p value (t-test), 95% CI, Mood's median test p Value |
|---|---|---|---|---|---|
| Cataract | 350 P or PS | 592, 472–738 (631, ±223) 206–1700 | 537, 435–692 (586, ±210) 206–1700 | 639, 540–785 (689, ±225) 310–1615 | **103, <0.0001, 66 to 140 <0.001** |
| Squint | 20 P or PS | 34, 23–47 (42, ±34) 2–304 | 32, 23–49 (43, ±37) 2–304 | 31, 23–46 (41, ±31) 5–197 | −2.03, 0.49, −7.8 to 3.7, 0.406 |
| Corneal grafts | 6 A | 9, 7–14 (12, ±10) 0–89 | 9, 6–13 (11, ±8) 0–66 | 10, 7–16 (14, ±11) 0–89 | **2.73, 0.001, 1.1 to 4.4, 0.015** |
| Corneal grafts | 0 P or PS | 0, 0–4 (6, ±15) 0–147 | 1, 0–5 (6, ±14) 0–147 | 0, 0–4 (6, ±16) 0–114 | 0.36, 0.7799, −2.18 to 2.89, 0.399 |
| Oculoplastic and lacrimal (excluding ptosis) | 40 P or PS | 116, 72–197 (158, ±133) 10–1010 | 97, 61–163 (138, ±124) 10–894 | 142, 89–231 (183, ±140) 39–1010 | **45.4, 0.0001, 23 to 68, <0.001** |
| Ptosis | 3 A | 9, 4–15 (11, ±11) 0–114 | 6, 3–13 (10, ±11) 0–58 | 10, 5–18 (13, ±11) 0–114 | **2.59, 0.0063, 0.73 to 4.45, <0.001** |
| Procedures for glaucoma (including laser) | 30 P or PS | 57 (80; ±63.1) 17–530 | 51 (72; ±58.6) 17–530 | 70 (91; ±67.1) 23–402 | **19, 0.0006, −8.2 to 29.9, <0.001** |
| Retinal detachment and VR | 20 A | 34, 22–56 (49, ±47) 0–435 | 29, 21–51 (41, ±38) 0–356 | 40, 25–74 (58, ±56) 0–435 | **17.23, 0.0001, 8.9 to 25.6, <0.001** |
| Retinal detachment and VR | 0 P or PS | 6, 1–23 (45, ±117); 0–1197 | 8, 2–25 (51, ±139) 0–1197 | 5, 1–21 (37, ±82) 0–467 | −13.11, 0.17, −31.9 to 5.71, 0.10 |
| Laser to retina | 40 P or PS | 160, 98–238 (185, ±125) 8–934 | 140, 84–222 (171, ±123) 8–934 | 176, 114–259 (203, ±125) 40–708 | **31.5, 0.0036, 10.3 to 52.7, 0.004** |

The final column shows the difference in means between the mean number of surgical procedures performed by OST and HST individuals. Statistically significant differences (p<0.05) are given in bold.

A, assisted; CCT, Certificate of Completion of Training; HST, higher surgical training; OST, ophthalmic specialist training; P, performed; PS, performed supervised; VR, vitreoretinal.

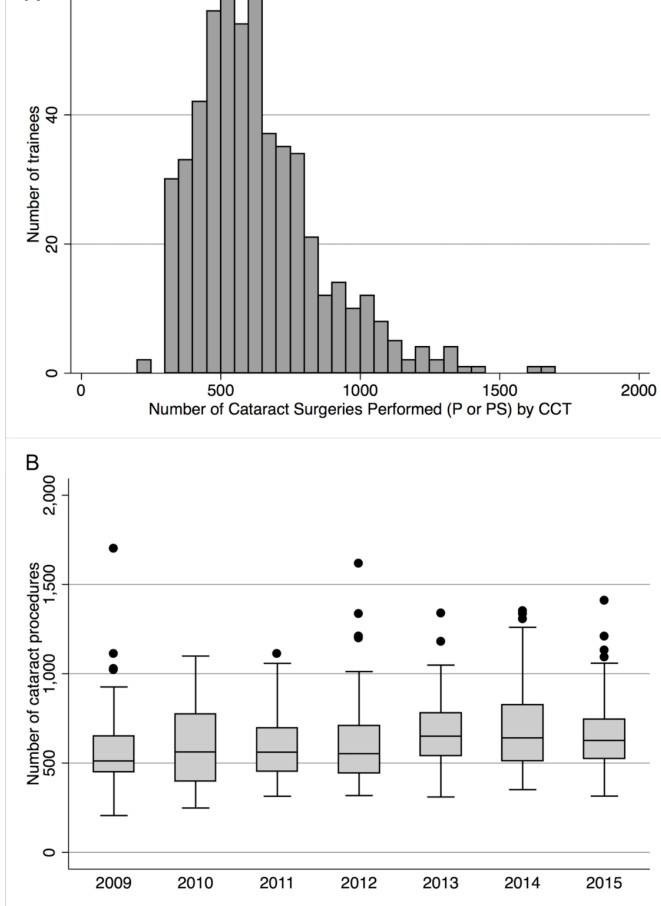

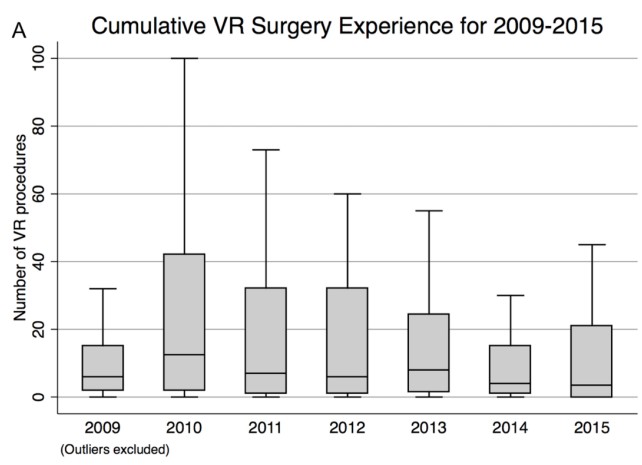

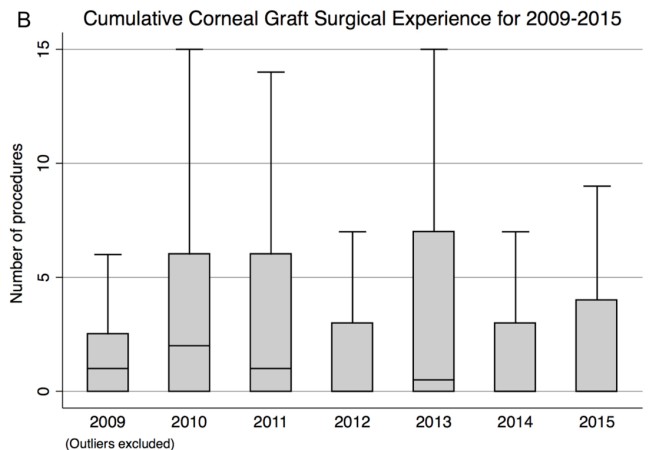

**Figure 2** (A) Frequency histogram showing total number of cataract surgeries performed by trainees at the time of accreditation (Certificate of Completion of Surgical Training or Certificate of Completion of Training (CCT)) during our 7-year study period (2009–2015); (B) Cumulative cataract surgery experience for 2009–2015. P, performed; PS, performed supervised.

**Figure 3** (A) Cumulative vitreoretinal (VR) surgery experience ('performed/performed supervised' (P/PS)) for 2009–2015; (B) cumulative corneal graft surgery experience (P/PS) for 2009–2015. Note the outliers have been omitted to allow for scaling.

among trainees is also greater in the UK compared with other countries, including Germany and Australia.[16 17]

For squint, corneal surgery and VR surgery, numbers of cases performed appeared stable over the study period, though somewhat lower when compared with the previous analysis by Ezra *et al*. For squint surgery, Ezra *et al*, noted a 'large downward trend' in surgery numbers, from median numbers of 121 in 1993 to 43 in 2008.[13] For VR surgery, median case numbers were typically 45–50 cases performed in the mid-1990s, compared with median case numbers ranging between 4 in 2015 and 13 in 2010. Similarly, for corneal grafts there was a significant decrease with a trend towards a trainee having assisted only and not performed any corneal graft surgery at time of CCT. This truncation of surgical numbers may be because of the curricular requirement throughout the period being only to have assisted at corneal graft surgery and to have performed only 20 strabismus procedures. However, there were considerable outliers over our 2009–2015 study period, with some trainees having recorded

performing over 1000 VR procedures by their time of CCT. This extraordinary variability may reflect different interpretation of how procedures are recorded, or which procedures are recorded under each category. In VR, for example, it may be that this represents large numbers of intravitreal injections performed by some trainees. For oculoplastics there was a trend towards increasing numbers in line with that noted previously and is likely due to the definition of oculoplastic procedures being widened in 2003 by the RCOphth to include 'minor op' procedures such as incision and curettage of chalazia.[13] Glaucoma procedures have slightly increased over the study period. Glaucoma experience was not reported by Ezra *et al* and so it is difficult to comment on whether this is a long-term trend. It is important to note, however, that these procedures include laser procedures such as cyclodiode and peripheral iridotomies where surgical exposure is limited, and so it is impossible to comment on the true glaucoma surgical experience.

This is the first study to report cumulative surgical experience from ophthalmology trainees since the implementation of MMC and run-through training. Rodrigues *et*

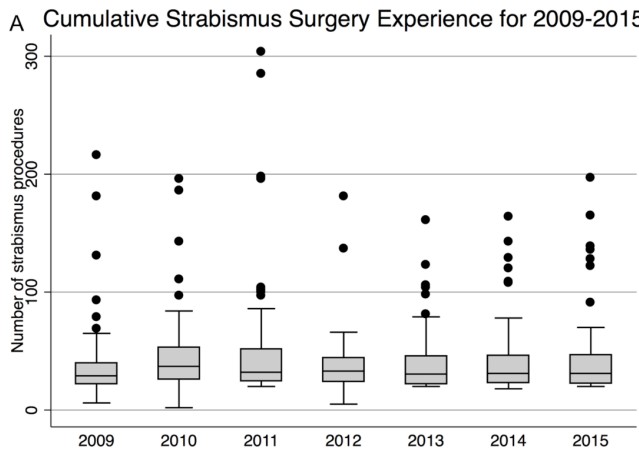

A Cumulative Strabismus Surgery Experience for 2009-2015

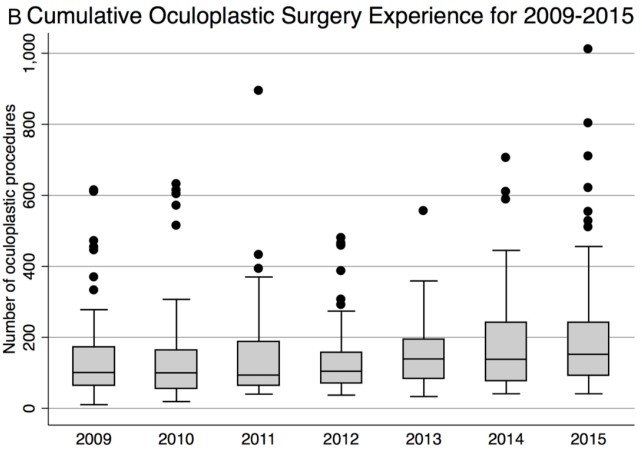

B Cumulative Oculoplastic Surgery Experience for 2009-2015

**Figure 4** (A) Cumulative squint surgery experience ('performed/performed supervised' (P/PS)) for 2009–2015; (B) cumulative oculoplastic surgery experience (P/PS) for 2009–2015.

al conducted a cross-sectional survey of ophthalmology trainees at year 3 level or above on the new MMC-based training scheme in 2012.[18] They raised a concern that cataract numbers were possibly in decline since MMC and

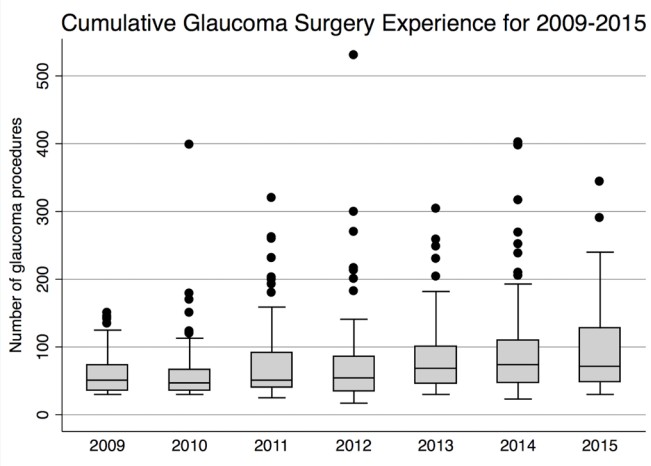

**Figure 5** Cumulative glaucoma surgical experience ('performed/performed supervised') for 2009–2015.

attributed this to the reduction in working hours. The estimated average number of hours worked by trainees prior to MMC was reported as 30 000 hours, which is considerably more than the 9000 hours of work-based experiential learning obtained during the 7 years of OST.[8 19 20] Of note, our analysis found OST trainees reported approximately 100 more cataract surgical cases than HST trainees, and for several areas OST trainee case numbers were higher than HST case numbers, as seen in table 1. However, the HST data does not include any case numbers performed during BST prior to HST entry. A HST trainee would typically have spent 2–4 years as an SHO (BST) before competitive entry to the HST programme (4.5 years), and, for example, a trainee would normally have had to be proficient in cataract surgery for entry to HST. It is therefore expected that OST trainees may have overall performed fewer surgical cases than HSTs at the time of CCT. This is of particular concern for subspeciality procedures such as squint surgery where HST and OST reported case numbers were similar, rather than proportionately higher as one might expect for a 7 year versus 4.5-year training programme. This may well be explained by the current emphasis on a shift towards training general ophthalmologists.[9 21] The requirement to train comprehensive general ophthalmologists will continue with the recently published Shape of Training report, which recommends that instead of focused specialty training, trainees of the future should train in a 'broad-based' manner to gain experience within a particular theme, thereby allowing easier workforce management and a clear shift to work in the community.[22] Our data are in part in agreement with this suggesting subspecialty surgical experience is in decline.[13 18]

There are several limitations to our study. Case numbers are self-reported from trainees' logbooks. However, these are verified at each year of training by a named educational supervisor and the cumulative numbers at CCT by the local training programme director. This can also be audited by the Royal College. It does, however, largely depend on the honesty and integrity of the trainee to accurately update their logbooks and the way that they interpret how cases should be recorded; for example, intravitreal injections being recorded as VR procedures. Additionally there have been advances in surgical training, particularly with regards to simulation with mounting evidence as to their improvement of surgical skills,[23] and some simulated cases can now be counted towards competency attainment. There is expected to be some under-reporting: all trainees must demonstrate that they have assisted at six corneal grafts yet some trainees reported not assisting in corneal graft cases (a curricular requirement) but instead reported performing more than six as the primary surgeon (P or PS). Surgical numbers and logbooks only tell part of the story of an individual trainee's surgical experience. Indeed, 'The procedure list is a crude surrogate for the assessment of technical skill. Its only redeeming quality is its ability to assure the depth and breadth of a resident's operative experience'.[24]

This is a retrospective study and the individual time in training has not been taken into consideration and may be heterogeneous, particularly for trainees prior to MMC and those who have had atypical training pathways. It does not consider additional other surgical experience, that is, detailed in trainees' e-portfolios which also form part of the CCT requirements, such as removal of eye or temporal artery biopsies. Nor does it consider the complication rates of surgical procedures, the ability of trainees to deal with complications or the 'readiness' of a trainee for a consultant post. Finally, this study does not assess regional variation in surgical numbers as there may be considerable variation in opportunities and training between deaneries.[18]

Despite the limitation of the working week to 48 hours, introduction of OST and associated phase out of HST, cataract surgical experience and many subspeciality operation numbers appear on initial inspection to have remained constant or with a marginal trend towards increasing case numbers. HST (4.5-year programme) case numbers do not include those performed before entry to HST, and although case numbers tended to be higher for OST trainees (7-year programme) compared with HST trainees, they were not proportionately so as would be expected. Looking ahead, postgraduate medical and surgical training will be under further pressure. There will likely be further restriction on the working week due to more out-of-hours emergency work by virtue of the new contract imposed by NHS Employers,[25] and a possible further shift to even more generalist training.[22]

**Contributors** JH: data collection, statistical analysis, database management, wrote first draft of manuscript, wrote further drafts of paper, submitted paper. DE: reviewed first and subsequent drafts of manuscript, and provided feedback. FS: reviewed first and subsequent drafts of manuscript, and provided feedback. ACD: devised study, supervised research, edited first and subsequent drafts of paper, correspondence with Royal College and other stakeholders.

**Funding** ACD and JH were supported by the National Institute for Health Research (NIHR) Biomedical Research Centre based at Moorfields Eye Hospital NHS Foundation Trust and UCL Institute of Ophthalmology.

**Disclaimer** The views expressed are those of the author(s) and not necessarily those of the National Health Service, the NIHR or the Department of Health.

**Competing interests** None declared.

**Provenance and peer review** Not commissioned; externally peer reviewed.

**Data sharing statement** No additional unpublished data are available for this study.

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
