## [Reviewer comments · BMJ Open]

ARTICLE DETAILS

TITLE (PROVISIONAL)	Changes in UK Ophthalmology Surgical Training: Analysis of Cumulative Surgical Experience 2009-2015
AUTHORS	Hoffman, Jeremy; Spencer, Fiona; Ezra, Daniel; Day, Alexander

VERSION 1 - REVIEW

REVIEWER	Ian Rodrigues St Thomas' Hospital, London
REVIEW RETURNED	16-Jul-2017

GENERAL COMMENTS	Overall, a well-written paper that answers an important and relevant question. A limited data set, but appropriately analysed and results reported clearly. Comprehensive enough and well-balanced discussion. Specific comments: Page 3, line 70 - although surgical numbers are "self-reported", it should be stated that the self-reported numbers are verified at each year of training by a named educational supervisor and cumulative numbers are validated by the regional training programme director Page 4, line 87 - was the SpR training duration following the Calman report not 4.5 years rather than 72 months? Page 4, line 90 - I would say that ophthalmology in the UK still operates an "on-call" rather than a "shift" system Page 4, line 92 - "statutory enforcement" which you can opt out of Page 4, line 94 - although MMC was supposed to be competency based, there is definitely still a time-based with a requirement of a minimum of 72 months of accredited ophthalmology training (unless application for accelerated training has been approved on an individual basis) Table 1, final column - is the difference in means between OST training numbers and the previous system? The legend or column heading should be amended to clarify Figure 3 - it should be stated more clearly if these numbers represent numbers performed Page 10, line 273-277 - see previous comment regarding validation of logbooks Page 11, line 289 - could also add that complication rates, competency in dealing with complications are also not considered.
---

	Neither is the "readiness" of trainees for taking a consultant job or how much further training if any is undertaken before commencing a substantive consultant post
--	--

REVIEWER	Nisha Chadha Icahn School of Medicine at Mount Sinai
REVIEW RETURNED	05-Aug-2017

GENERAL COMMENTS	This is interesting manuscript which reports trends in UK trainee surgical experience in the setting of duty hour restriction and change in training scheme. It contributes relevant information to ophthalmology educators. A major limitation of the data is that it was self reported by trainees and and that laser/minor procedures are incorporated into the surgical numbers. However, the authors do address this limitation. It would be helpful if the authors elaborated a bit more on the UK system of education and what experiences are involved in BST vs. HST vs. OST. It would also be informative to describe the subspecialty ophthalmology training experience and duration, especially given the manuscript suggests that stress will be placed on learning subspecialty procedures as a part of this higher training.
---

VERSION 1 – AUTHOR RESPONSE

Reviewer 1:

Overall, a well-written paper that answers an important and relevant question. A limited data set, but appropriately analysed and results reported clearly. Comprehensive enough and well-balanced discussion.

Thank you.

Specific comments:

Page 3, line 70 - although surgical numbers are "self-reported", it should be stated that the self-reported numbers are verified at each year of training by a named educational supervisor and cumulative numbers are validated by the regional training programme director

This has been stated in the text, lines 71-74

Page 4, line 87 - was the SpR training duration following the Calman report not 4.5 years rather than 72 months?

Thank you, I have corrected this in the text (line 92)

Page 4, line 90 - I would say that ophthalmology in the UK still operates an "on-call" rather than a "shift" system

This has been stated in the text, lines 95-97

Page 4, line 92 - "statutory enforcement" which you can opt out of

This has been stated in the text, line 99

Page 4, line 94 - although MMC was supposed to be competency based, there is definitely still a time-based with a requirement of a minimum of 72 months of accredited ophthalmology training (unless application for accelerated training has been approved on an individual basis)

This has been stated in the text, lines 102-104

Table 1, final column - is the difference in means between OST training numbers and the previous system? The legend or column heading should be amended to clarify

This has been clarified in the legend of Table 1

Figure 3 - it should be stated more clearly if these numbers represent numbers performed

The legends for the graphs have been amended accordingly to make this clearer.

Page 10, line 273-277 - see previous comment regarding validation of logbooks

This has been stated in the text, lines 301-303

Page 11, line 289 - could also add that complication rates, competency in dealing with complications are also not considered. Neither is the "readiness" of trainees for taking a consultant job or how much further training if any is undertaken before commencing a substantive consultant post

This has been stated in the text, lines 326-327

Reviewer 2:

This is interesting manuscript which reports trends in UK trainee surgical experience in the setting of duty hour restriction and change in training scheme. It contributes relevant information to ophthalmology educators. A major limitation of the data is that it was self reported by trainees and that laser/minor procedures are incorporated into the surgical numbers. However, the authors do address this limitation.

Thank you for your comments. As you have mentioned, we have mentioned these limitations in the discussion.

It would be helpful if the authors elaborated a bit more on the UK system of education and what experiences are involved in BST vs. HST vs. OST. It would also be informative to describe the subspecialty ophthalmology training experience and duration, especially given the manuscript suggests that stress will be placed on learning subspecialty procedures as a part of this higher training.

I have elaborated on the roles of HST, BST and OST trainees, as well as on the subspecialty experience. Please see lines 112-118 and lines 123 - 130 respectively.

I hope these changes are satisfactory and I look forward to hearing your thoughts in due course. If there are any further changes to be made, please do not hesitate to contact me.